# Neurologic Injury-Related Predisposing Factors of Post-Traumatic Stress Disorder: A Critical Examination

**DOI:** 10.3390/biomedicines11102732

**Published:** 2023-10-09

**Authors:** Wiley Gillam, Nikhil Godbole, Shourya Sangam, Alyssa DeTommaso, Marco Foreman, Brandon Lucke-Wold

**Affiliations:** 1College of Medicine, University of Florida, Gainesville, FL 32610, USA; gillamw@ufl.edu (W.G.);; 2School of Medicine, Tulane University, New Orleans, LA 70112, USA; ngodbole@tulane.edu; 3College of Liberal Arts and Sciences, University of Florida, Gainesville, FL 32603, USA; 4College of Health Professions and Sciences, University of Central Florida, Orlando, FL 32827, USA; 5Department of Neurosurgery, University of Florida, Gainesville, FL 32610, USA

**Keywords:** Post-Traumatic Stress Disorder, neurologic injury, traumatic brain injury, subarachnoid hemorrhage, stroke

## Abstract

The present review aimed to identify the means through which neurologic injury can predispose individuals to Post-Traumatic Stress Disorder (PTSD). In recent years, comprehensive studies have helped to clarify which structures in the central nervous system can lead to distinct PTSD symptoms—namely, dissociative reactions or flashbacks—when damaged. Our review narrowed its focus to three common neurologic injuries, traumatic brain injury (TBI), subarachnoid hemorrhage (SAH), and stroke. We found that in each of the three cases, individuals may be at an increased risk of developing PTSD symptoms. Beyond discussing the potential mechanisms by which neurotrauma may lead to PTSD, we summarized our current understanding of the pathophysiology of the disorder and discussed predicted associations between the limbic system and PTSD. In particular, the effect of noradrenergic neuromodulatory signaling on the hypothalamic pituitary adrenal (HPA) axis as it pertains to fear memory recall needs to be further explored to better understand its effects on limbic structures in PTSD patients. At present, altered limbic activity can be found in both neurotrauma and PTSD patients, suggesting a potential causative link. Particularly, changes in the function of the limbic system may be associated with characteristic symptoms of PTSD such as intrusive memories and acute psychological distress. Despite evidence demonstrating the correlation between neurotrauma and PTSD, a lack of PTSD prognosis exists in TBI, SAH, and stroke patients who could benefit from early treatment. It should be noted that PTSD symptoms often compound with pre-existing issues, further deteriorating health outcomes for these patients. It is ultimately our goal to clarify the relationship between neurotrauma and PTSD so that earlier diagnoses and appropriate treatment are observed in clinic.

## 1. Introduction: Neurotrauma Preceding Post-Traumatic Stress Disorder

PTSD is characterized by a wide-ranging set of symptoms including but not limited to intrusive and distressing memories, dissociative reactions, and serious psychological distress in response to stimuli that resemble a previously experienced traumatic event [1]. Common to all individuals diagnosed with PTSD is exposure to trauma. However, this trauma may be physical or emotional, suggesting that the disorder does not require neurologic injury. Though it may not be inherent in all cases, many studies indicate a correlation between traumatic brain injury (TBI) and PTSD [2,3,4,5,6]. One meta-analysis found a 13.5% prevalence rate of PTSD in civilians who had been diagnosed with mild traumatic brain injury (mTBI) [2]. Another study found that greater than 17% of United States Army soldiers with mTBI returning from Operation Enduring Freedom (OEF) in Afghanistan and Operation Iraqi Freedom (OIF) in Iraq screened positive for PTSD [4].

In addition to TBI, stroke and subarachnoid hemorrhage may predispose one to PTSD. Several studies document the development of PTSD symptoms in individuals following stroke [7,8,9]. These symptoms can be assessed using several measures. The present review focuses primarily on studies which utilized the gold-standard technique, the Clinician-Administered PTSD Scale (CAPS), as found in the Diagnostic and Statistical Manual of Mental Disorders, Fifth Edition (DSM-V) [7,10]. A significant proportion of stroke and subarachnoid hemorrhage patients present with PTSD symptoms a year following the event, suggesting PTSD treatment may prove beneficial for such individuals [7,10]. It should be noted that some of the studies included in the present review used DSM-IV which, arguably, has a less restrictive concept of PTSD. One significant discrepancy can be found where DSM-V defines trauma, thus precluding any subjective interpretations of the type of events—actual or threatened death, sexual violence, or serious injury—which must precede diagnosis [11].

Although it appears evident that neurotrauma can be involved in the development of PTSD in patients, it is unclear how the extent of an injury affects symptoms. One study claims that PTSD severity scales with the level of involvement in combat events, indicating the possibility of a dosage-dependent relationship between trauma exposure and the disorder [2]. An alternative explanation is that increased exposure to combat events increases the likelihood of a more serious neurologic injury, thereby worsening PTSD symptoms. However, there is little evidence for this theory. In fact, some data suggest that mTBI is more predictive of PTSD symptoms than moderate-to-severe TBI [12]. There are two important considerations to make when trying to interpret this result: (1) the sample sizes for most studies which have drawn this conclusion have been quite small and may not accurately represent the large population of patients suffering from both TBI and PTSD; (2) moderate-to-severe TBI is associated with longer durations of post-traumatic amnesia, which is inversely related to intrusive memories due to limited trauma memory encoding. Together, these factors further complicate our current understanding of how neurotrauma may be linked to PTSD.

## 2. The Pathophysiology of Post-Traumatic Stress Disorder

The major brain structures which have been proved to play some role in PTSD symptoms include the prefrontal cortex (PFC), hippocampus, amygdala, and other downstream elements such as the locus coeruleus (LC) and hypothalamus [13,14,15]. A critical component of the prefrontal cortex negatively regulates amygdala function to prevent hyperactivity. It has been found that PTSD patients exhibit reduced activity in the medial prefrontal cortex (mPFC) and anterior cingulate cortex (ACC) during presentation of trauma-related and non-related aversive stimuli [16,17]. Further, they have been observed to have decreased volume in the ventromedial prefrontal cortex (vmPFC) and ACC [16,17,18,19,20] and even to utilize altered pathways from the ACC to the amygdala [21]. All these changes in the PFC suggest that amygdala dysregulation may be involved in the pathophysiology of PTSD by erasure of the conventionally strict regulation of fear responses [13].

Noradrenaline is a neuromodulatory neurotransmitter system capable of producing a rapid and coordinated response to acute stress. A meta-analysis compiling data from 1388 articles along with several other studies found statistically significant increased levels of noradrenaline in PTSD patients compared to controls, indicating the likelihood of increased noradrenergic tone in these individuals [22,23,24,25]. Of note, one study conducted in 2001 observed concentrations of NA in hourly CSF samples (collected over a 6 h period) to be higher in male combat veterans compared to healthy controls [26]. Based on these findings, Pietrzak and colleagues conducted a follow-up study to determine whether a reduction in the Norepinephrine Transporter (NET) could be responsible for the increased noradrenergic tone in PTSD patients [27]. The researchers found evidence for a significant reduction in the density of NET labeled in the LC in PTSD patients compared to controls [27]. This was a profound discovery that seemingly identified a specific neural correlate of PTSD in combat veterans. However, it is important to note that this article has since been retracted. As such, further research is still needed to determine the legitimacy of these findings.

As previously mentioned, changes in the PFC found in PTSD patients have downstream impacts on the amygdala. Specifically, it has been found that the amygdala in those with PTSD is hyperactive during exposure to trauma-linked events [28]. A critical function of the amygdala is to stimulate the hippocampus during the formation of new memories related to fear-inducing events. A lack of proper control of the amygdala provides a possible explanation as to why PTSD patients experience intrusive memories from prior trauma. Although the results were similar for both trauma-exposed individuals and PTSD patients, meta-analyses observed statistically significant differences between these two groups and healthy controls [29]. Reduced cortisol levels and signaling in PTSD patients could be responsible for stronger sympathetic nervous system activation and extreme lucidity of traumatic memories [30,31,32,33]. Notably, Robert Sapolsky claimed that, under stress, the body releases glucocorticoids, resulting in downregulation of glucocorticoid receptors, thereafter, leading to hippocampal atrophy in response to chronic stress [34]. Only a decade later, he described the relevance this theory has to PTSD, whereby hyperactivation of this stress system in response to trauma could lead to cell death in the hippocampus and eventually cognitive dysfunction [35]. Particularly compelling evidence for hippocampal involvement in PTSD was found by several meta-analyses reporting a significant reduction in hippocampal volume in PTSD patients as compared to both healthy controls and trauma-exposed individuals without PTSD [36,37].

Norepinephrine and indirect signaling from the limbic system including the hippocampus, mPFC, and amygdala act on the paraventricular nucleus of the hypothalamus to trigger the release of corticotropin-releasing hormone, or CRH [38]. The HPA axis which is initiated by CRH release is the single most critical hormonal facilitator of stress response in humans. Rodent models have demonstrated that PTSD-associated behaviors are observed in response to CRH injection [39] and that CRH receptor binding in the amygdala can produce fear responses [40]. Despite substantial evidence implicating CRH in high-stress states and PTSD symptoms, results from other studies complicate the matter. Notably, CRH release in the bed nucleus of the stria terminalis has an inhibitory effect on neurons in the amygdala, which results in a decreased fear response [41]. This clearly contradicts the idea that CRH release necessitates amygdala excitation. It is possible that CRH differentially affects the amygdala according to the pathway it uses, but further evidence is needed to confirm this theory.

Several studies have also looked downstream of CRH to determine whether cortisol is involved in the development of PTSD. It has been found that lower concentrations of hair cortisol in military veterans prior to overseas deployment was positively correlated with PTSD symptoms post deployment [42]. Likewise, police academy recruits with reduced increases in cortisol while watching videos of traumatic situations were more likely to demonstrate signs of long-term distress and decreased resilience after four years [43]. Together, these findings seem to indicate that lower cortisol concentrations predict PTSD symptoms following exposure to trauma. However, a meta-analysis demonstrated that heart rate was the only biological measure obtained post trauma that could predict PTSD [44]. Additionally, even if low cortisol concentrations are associated with PTSD, no threshold cortisol concentration has been identified, leaving little clinical applicability. It is evident that more research is needed to clarify the relationship between the HPA axis and PTSD. Figure 1 is a greatly simplified model that serves to clarify the complex pathophysiology of PTSD, but it should be noted that the relationship between these neurological structures and PTSD is still widely contested.

## 3. Traumatic Brain Injury

Traumatic brain injury (TBI) is defined as the alteration of brain physiology and function secondary to an external force [45]. TBIs can range from mild, often referred to as “concussions”, to moderate and severe forms, each bearing its distinguishing features and implications [46]. TBI has emerged as a significant area of investigation with respect to understanding its role in rendering individuals more susceptible to developing PTSD. Studies exploring the connection between TBI and PTSD in different populations have provided valuable insights into how TBI could affect the emergence and comorbidity of PTSD symptoms. A study published in 2008 by Hoge et al. found that mTBI experienced by soldiers deployed in Iraq was strongly associated with PTSD and physical health problems. This study suggests that addressing PTSD is critical for improving the physical and psychological well-being of soldiers with TBI [47]. Another study found that when black women experienced TBI from intimate partner violence, there was a subsequent increase in the presence of both PTSD and depression. After accounting for other factors, the statistical analysis determined that the occurrence of TBI was associated with an increase in comorbid PTSD and depression symptoms [48]. In terms of children and adolescents, one study found that, after experiencing a severe TBI, the percentage of those with PTSD increased to 13%, which is significantly higher than the general child population’s lifetime prevalence of 7.8%. This difference in PTSD prevalence between children with severe TBI and those without suggests a strong correlation between TBI and the subsequent development of PTSD in children and adolescents [49].

Several potential mechanisms could explain why TBI might predispose different populations of individuals to PTSD. One theory is that TBI can lead to damage to the brain regions involved in emotion regulation, such as the amygdala and hippocampus, which are known to play a role in the development of PTSD [50]. TBI can damage the hippocampus, which may lead to impaired memory and emotion processing, both of which have been linked to PTSD [51]. Additionally, TBI has been associated with altered activity in the amygdala, a brain region involved in the recognition and processing of fear, which is thought to play a key role in the development of PTSD [52]. Moreover, TBI can lead to dysregulation of neurotransmitters such as serotonin and norepinephrine, which have also been implicated in the development of PTSD [53]. Lastly, TBI can also lead to damage to the frontal lobe, which is involved in emotional regulation and impulse control, which could potentially contribute to the development of PTSD [52]. More on these relationships can be found in Figure 2.

Expanding on the relationship between TBI and PTSD, another potential mechanism centers around how TBI can lead to feelings of vulnerability and powerlessness. The physical injury itself, resulting from a forceful impact on the head, can create a profound sense of helplessness in individuals who cannot care for themselves. This reliance on others for basic needs can exacerbate feelings of vulnerability and a loss of autonomy. TBI can also cause cognitive changes, which can disrupt a person’s sense of capability and self-assurance. This emotional response, originating from the very nature of the trauma, could establish a foundation that strengthens the risk of developing PTSD [54]. In military personnel who undergo combat-related trauma, the added layer of exposure to life-threatening situations and witnessing the traumatic experiences of others can heighten these feelings of vulnerability and powerlessness, strengthening the link between TBI and the susceptibility to PTSD. For these individuals, the combination of TBI and the psychological complications of combat stressors becomes a convoluted interplay that shapes their psychological response. Furthermore, people with a history of emotional and physical abuse may also be more susceptible to developing PTSD following TBI due to their pre-existing issues with emotional regulation and stress management [55]. This means that TBI may exacerbate already present risk factors, making it more likely for individuals to develop PTSD. In essence, what occurs is a dynamic where TBI not only influences cognitive aspects but also intersects with pre-existing vulnerabilities. By heightening feelings of vulnerability and exacerbating pre-existing emotional regulation issues, TBI can act as an accelerant for the development of PTSD.

## 4. Subarachnoid Hemorrhage

Subarachnoid hemorrhage (SAH) is a severe neurological condition that is characterized by spontaneous bleeding into the subarachnoid space, which consists of the cerebrospinal fluid, major blood vessels, and cisterns [56]. In 80% of cases of SAH without preceding trauma, this begins with the rupture of an intracranial aneurysm, with other causes including vascular malformations [57,58]. The major presenting symptom of SAH is the onset of a sudden, severe headache that reaches peak intensity within seconds [59]. The death rate of SAH is approximately 50%, and this includes up to 18% of all aneurysmal SAH patients who die at home or in transit to a hospital [60]. About half of the surviving patients suffer from long-term effects and experience a subsequent decreased quality of life due to a significant risk of developing physical, memory, and neurocognitive impairments [61,62,63].

Based on current clinical studies, the best predictor of a patient’s neurological outcome is the level of impairment and extent of subarachnoid bleeding at the time of hospital admission [63,64]. A study that followed 415 eligible patients with SAH over a 90-day period after hospital discharge in Vietnam concluded that the Prognosis on Admission of Aneurysmal Subarachnoid Hemorrhage grading scale was most reliable in predicting poor outcomes [61]. However, predicting the overall outcomes of SAH remains imperfect because of the high variability of the clinical conditions, complications, and treatments administered [65,66,67]. For example, in the aforementioned study, a number of the patients experienced complications such as pneumonia and rebleeding, which greatly contributed to a high rate of poor outcomes [61]. Nonetheless, grading SAH patients upon admission is crucial to predicting their short- and long-term health outcomes.

Emerging research findings suggest that survivors of SAH may be at higher risk of developing Post-Traumatic Stress Disorder (PTSD). The clinical symptoms and postictal events of aneurysmal SAH can be a significantly traumatic experience that leads to a PTSD diagnosis for some [68,69]. Figure 3 illustrates common health outcomes for patients following SAH, which can often co-occur. In a study that conducted PTSD assessments on 105 SAH patients at 3 and 13 months post-ictus, 37% of the patients met the requirements of a PTSD diagnosis during both assessment periods, which is four times the rate of PTSD in the general population [70]. The patients of this study also reported elevated levels of fatigue and increased problems with their sleep, both of which are linked to PTSD. However, this study failed to demonstrate a clear pattern of PTSD symptoms over the course of data collection. Existing models have consistently shown differences in PTSD symptoms when the disorder arises from a life-threatening medical event versus a past traumatic external event, such as combat [71]. Furthermore, demographic factors such as age, sex, educational level, and cognitive ability also affect individual PTSD symptoms [72,73,74,75,76]. The sociodemographic factors that have been associated with poor outcomes in SAH patients include advanced age, female sex, and race, specifically, being African American, American Indian, Alaskan Native, Asian, or a Pacific Islander rather than white [77,78,79]. A study that investigated the differences in outcomes between male and female SAH patients noted that women may have been more affected as they are more likely to have had a prior psychiatric history than men [80].

In a study that examined the specific PTSD symptoms of aneurysmal SAH patients over the course of one year following the event, a variety of PTSD symptom patterns was observed. Researchers noted: about 50% of the patients did not show symptoms suggesting PTSD at any point, 15% showed recovery following PTSD symptoms at the first assessment, 9% exhibited symptoms suggesting PTSD during all assessments, and 13% experienced delayed symptoms [81]. The study’s findings were, however, dependent on a self-report questionnaire used to evaluate PTSD symptoms (not as accurate as PTSD diagnosis by formal psychiatric review) and a multilevel analysis that could not completely explain differences in the course of PTSD [81]. Nevertheless, the finding that half of the patients did not experience any PTSD symptoms in the first year post SAH is consistent with similar studies of SAH patients [70,82,83].

These results reinforce that a majority of SAH patients do not initially show symptoms indicative of PTSD yet later receive a diagnosis. Thus, it is imperative to evaluate these individuals for PTSD symptoms throughout the first year following SAH diagnosis. Despite the increased availability of treatment options, up to 70% of individuals experiencing PTSD symptoms still do not pursue professional care for their condition [84]. Further research must be conducted to better understand which SAH patients are likely to experience chronic PTSD. Particularly, studies that can sufficiently account for factors such as coping style, psychiatric history, and levels of social support would help to better specify the differences in variation of PTSD symptoms over time.

## 5. Stroke

Ischemic and hemorrhagic strokes account for 11.8% of deaths worldwide [81]. Like PTSD, a stroke is unexpected, uncontrollable, and life threatening. A meta-analysis of 1138 participants suggested 25% of stroke survivors will develop post-traumatic stress symptoms, and one in nine will develop chronic PTSD [82]. Stroke-induced PTSD has been linked with worse long-term stroke outcomes, including recurrent stroke [82], greater disability, and nonadherence to medications [83]. This may result in poor compliance with medications intended for secondary prevention of recurrent stroke. Non-compliance with these medications may suggest that stroke-induced PTSD could be a novel risk factor for recurrent stroke. Stroke-associated PTSD is characterized by intrusive symptoms such as flashbacks of the stroke/transient ischemic attacks (TIA), persistent avoidance of stimuli associated with the stroke, altered arousal, increased irritability, and negative alterations in cognition and mood [84]. The literature suggests that the presence of these mental health disorders has a large impact on quality of life and functional outcomes after stroke. Stein et al. found that in a population of mild stroke survivors with no measurable differences in neurologic deficits, patients with PTSD had significantly reduced quality of life scores [85]. There is also evidence that post-stroke psychiatric conditions may increase healthcare utilization and costs [86]. Mohd Zulkilfly et al. found that in 147 stroke patients, the presence of post-traumatic stress symptoms was a significant predictor of functional disability [87].

Recent literature has made great strides in identifying pertinent risk factors (see Figure 4). These include psychiatric comorbidities, neurotic coping strategies, low socioeconomic status, female sex, recurrent stroke, and past trauma and disability [84]. Multiple reports have correlated pre-stroke anxiety and depression diagnosis with PTSD occurrence [88]. Current research suggests that the tendency to develop post-stroke PTSD may be linked to pre-stroke coping styles [89]. Individuals with repressor and low-anxious coping styles are less likely to develop PTSD compared to those with defensive and high-anxious coping styles [89]. Additionally, there may be increased prevalence in low socioeconomic populations, which have a propensity for recurrent strokes and poorer physical and mental health status [89,90]. Despite this trend, the literature lacks standardized consensus on the significance of individual socioeconomic predictors. A cross-sectional study assessing traumatic response post stroke found significant association in development of PTSD, particularly in females and individuals with comorbid anxiety disorder (80%) or depression (30–50%) [89]. This study did not find any associations with stroke severity [88]. The only significant association was the role of emotional experience and cognitive appraisal of the event itself [89]. To this end, the number of past traumatic events also conferred an exponential risk [91]. Number of strokes, negative affect, and cognitive appraisal were identified as significant predictors of the number and severity of PTSD symptoms [7]. Peritraumatic distress was also found to prospectively predict acute PTSD symptoms [7,92].

Currently, the pathophysiology of post-stoke PTSD remains poorly understood. Neuroimaging studies have implied selective brain atrophy may be implicated in PTSD pathophysiology. A prospective cohort study of first-ever TIA/post-stroke survivors found that probable PTSD development is linked with lower performance in cognitive and functional tests, as well as with pro-inflammatory states with elevated CRP. Stroke severity has also been associated with PTSD severity, specifically white matter lesion volume and hippocampal connectivity. Infarct volume and location have not. Lesions affecting connectivity are more likely to influence PTSD. Currently, there are no studies that have correlated stroke location with PTSD development. One study attempting to do this was inconclusive [93]. Although there is no established consensus on lesion localization and PTSD development, Rutovic et al. identified some suggestive relationships [94]. A higher incidence was found in patients with right-sided and brainstem strokes [94]. This was attributed to a tendency for delayed treatment and more pronounced disability [94]. However, again, this relationship does not distinguish between PTSD secondary to lesion location and that secondary to severe disability. Despite the absence of evidence correlating macroscopic findings, neuroendocrine dysregulation post stroke has been associated with PTSD development. Pitman et al. found that a cortisol deficit may be implicated in PTSD pathogenesis [95]. Abnormal glucocorticoid levels have been found to interfere with memory, learning, stress adaptation, and resilience [96]. Furthermore, low-dose hydrocortisone injections may serve to prevent PTSD in neurotrauma patients [97]. One caveat is that these findings are not specific to pathogenesis of PTSD post stroke. Further studies are needed in this area. Currently, metabolic associations between TIA/stroke and PTSD are unclear. Changes in N-acetylaspartate have been observed in both PTSD and in neural tissue fewer than three days after TIA [98]. Only correlative associations have been identified thus far.

Current targets for intervention include fostering social support (friend, spouse, partner) [89] and reducing social isolation. Other current management strategies include exposure therapy, which has demonstrated efficacy in treatment of combat veterans, and interventions to teach more effective coping skills [84]. In support of this, avoidance has been found to be the strongest predictor for post-traumatic stress symptoms [87]. This suggests that facing traumatic experiences by refining coping skills may have therapeutic benefits in this population. The literature on specifically treating PTSD in a post-stroke population is now showing some progress. Jiang et al. conducted a randomized controlled trial on 170 first-ever stroke patients and provided modest evidence that supportive psychological therapy can effectively treat post-stroke PTSD in its early stages [99]. A recent case report also showed clinically significant resolution of PTSD symptoms following three sessions of eye movement desensitization and reprocessing therapy [100]. This study was limited, however, due to assessment via self-report and its design as a case report. Current treatment efforts are focused on trialing existing PTSD therapies in a post-stroke population, but there is scant existing data for this. Therapies used in post-trauma PTSD should be evaluated in randomized controlled trials within the post-stroke population as well. Potentially effective medications include fluoxetine, paroxetine, sertraline, and venlafaxine [101]. Emerging treatment options include 3,4 methylenedioxymethamphetamine-assisted psychotherapy, virtual reality exposure therapy, ketamine, cannabinoids, prazosin, and repetitive transcranial magnetic stimulation [101]. Although not yet investigated in the post-stroke PTSD population, CBT has also been shown to be effective in the treatment of post-stroke mood disorders [85]. Additionally, traditional CBT may be challenging for stroke patients who have impaired mobility. To this end, the efficacy of telehealth therapy should be further researched.

The predominant limitation to the current data of the epidemiology is their reliance on questionnaires and scales, which tend to overdiagnose PTSD, as opposed to formal psychiatric interviews [84]. The development of PTSD post stroke is also confounded by the disability conferred by severe stroke, which may elicit a stress response of its own.

## 6. Discussion

Unlike previous reviews which have mostly been broad in scope or focused solely on associations of mTBI with PTSD, we focused on three specific types of neurotrauma (TBI, SAH, and stroke) which seem to have the highest comorbidity with PTSD. In doing so, we intended to illuminate potential mechanisms through which PTSD symptoms may develop following physical trauma. One potential limitation of the present review includes the use of “older” (published >10 years ago) manuscripts which may have implemented outdated techniques to obtain their data. To mitigate this, we excluded any older studies that had results inconsistent with current data. Another limiting factor of the study was the absence of a standardized protocol with explicit, systematic, and pre-determined methods of article selection. This could make the results of this review more difficult to reproduce. However, data supplied by the study were cross-referenced with several additional sources, and discrepancies were noted. In sum, the pathophysiology of PTSD typically involves dysregulation of the limbic system, namely the amygdala and hippocampus [13,35]. Complicating the development of pharmacological treatment for PTSD are the many upstream elements which can result in PTSD symptoms if damaged or destroyed. A key point is that even when activity levels in the limbic system are normal, trauma to downstream elements such as the hypothalamus can also contribute to PTSD by altering CRH and cortisol levels, which are both major actors in the body’s major stress responses [42,43]. Although external traumatic experiences such as combat are often attributed as a prerequisite for PTSD, internal medical events such as TBI, subarachnoid hemorrhage, and stroke are also associated with the psychiatric disorder. Furthermore, despite being reported by several studies that the majority of TBI, SAH, and stroke patients do not develop PTSD symptoms, these instances of neurotrauma are each still moderately predictive of PTSD, with an average rate of symptom development of 11–20% in TBI patients, 37% in SAH patients, and 25% in stroke patients [2,70,82]. Given these results and the importance of early diagnosis in psychiatric disorders, we believe changes in clinical procedures should be made to routinely screen for PTSD in the first year post trauma in all patients who have suffered neurologic injury, particularly TBI, SAH, or stroke.

TBI, SAH, and stroke may each be associated with a distinct pathogenesis of PTSD. The primary means described in TBI and SAH studies include damage to the amygdala and hippocampus, which affect emotional processing and memory, respectively [50,51,102]. Additionally, TBI has been observed to cause dysregulation of mood-altering neurotransmitters such as serotonin and norepinephrine, which may induce PTSD symptoms and explain comorbidities such as Major Depressive Disorder [53]. The correlation between stroke and PTSD is still ambiguous. There is some evidence for hippocampal connectivity being affected in stroke patients who later receive a PTSD diagnosis. Other evidence suggests abnormal glucocorticoid levels following stroke can induce PTSD, but all current evidence is correlative [95,96].

Through various means, it is likely that traumatic injury to the central nervous system, particularly the limbic system, acts to exacerbate fear and intrusive memories. These findings support the idea that many neurotrauma patients may benefit from regular PTSD screening to prevent the worsening of symptoms if left untreated. Future studies should aim to clarify the impacts of altered noradrenergic signaling on the HPA axis and its potential interplay with modified amygdala activity. It is possible that hyperactivity in the amygdala and robust CRH and cortisol responses to noradrenergic signaling may induce unusual stress behaviors such as those seen in PTSD patients. Developing a concrete understanding of the physiological underpinnings of PTSD may allow for differential treatment of those who previously suffered neurotrauma compared to those who have not.

## 7. Conclusions

Neurologic injury in the form of TBI, SAH, and stroke is associated with increased risk of development of PTSD symptoms. The primary mechanism by which neurotrauma induces PTSD is likely associated with modified activity in limbic structures, namely the amygdala.

## Figures and Tables

**Figure 1 biomedicines-11-02732-f001:**
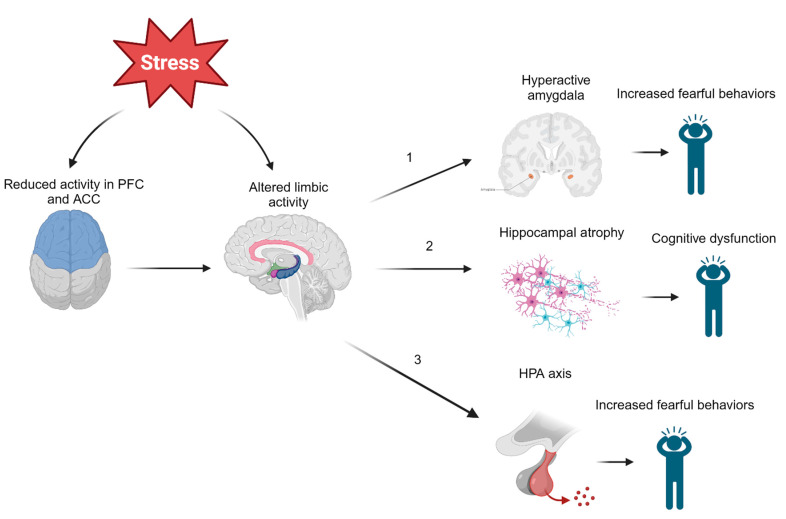
Effects of Modulated Limbic Activity. Chronic or high acute stress experienced during and after trauma may directly and indirectly act on several limbic structures. (1) Altered regulation of the amygdala may induce fearful behaviors observed in those with PTSD. (2) Stress may also lead to hippocampal atrophy, which could be responsible for many symptoms of PTSD, including cognitive dysfunction, involuntary recurrent memories (flashbacks), and more. (3) Limbic structures such as the hippocampus or amygdala may act on the hypothalamus to activate the HPA axis by CRH release. These neuroendocrine disruptions in the body’s primary stress response mechanism could also induce fearful behaviors. Note: ACC = anterior cingulate cortex, PFC = prefrontal cortex, HPA = hypothalamic–pituitary–adrenal axis.

**Figure 2 biomedicines-11-02732-f002:**
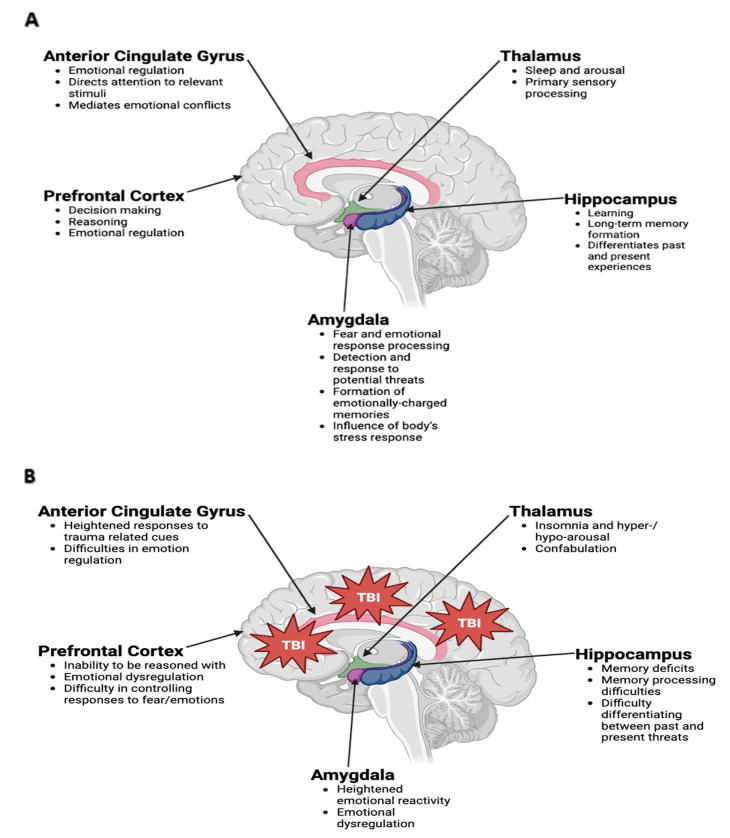
Limbic System. (**A**) Anatomical representation of the limbic system, which is composed of delicate neuronal structures including the anterior cingulate gyrus, thalamus, hippocampus, amygdala, and prefrontal cortex, among others. The limbic system plays vital roles related to memory, learning, and behavior through a complex interplay among the aforementioned structures. (**B**) Representation of these structures and their subsequent dysregulation following traumatic brain injury (TBI), which can be delivered in the form of external force and/or psychological insult.

**Figure 3 biomedicines-11-02732-f003:**
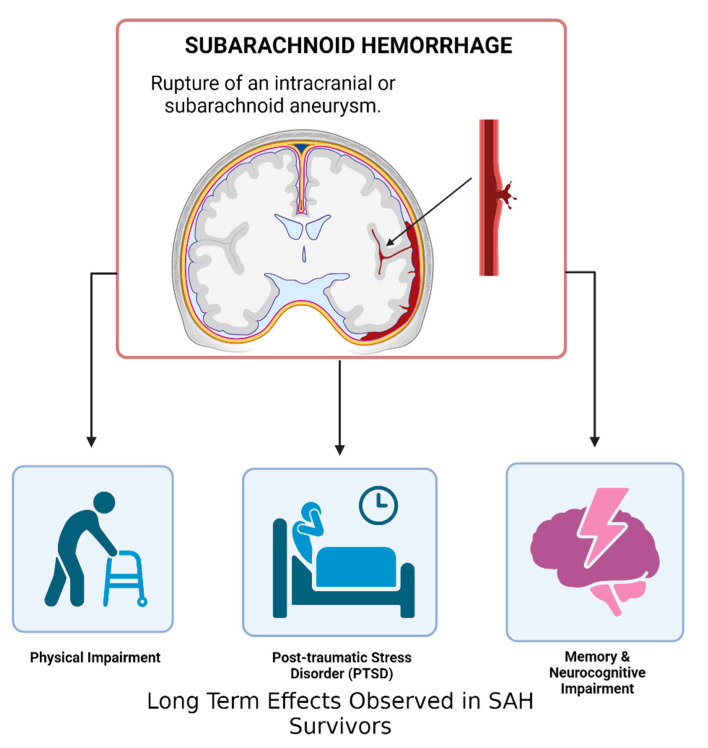
Long Term Effects of Subarachnoid Hemorrhage. Subarachnoid hemorrhage (SAH) is caused in most cases by the rupture of an intracranial aneurysm, resulting in bleeding into the subarachnoid space (the space between the arachnoid mater and pia mater). Of SAH patients, 50% survive, of which half suffer from long-term effects such as physical, memory, and neurocognitive impairment as well as potential PTSD symptoms.

**Figure 4 biomedicines-11-02732-f004:**
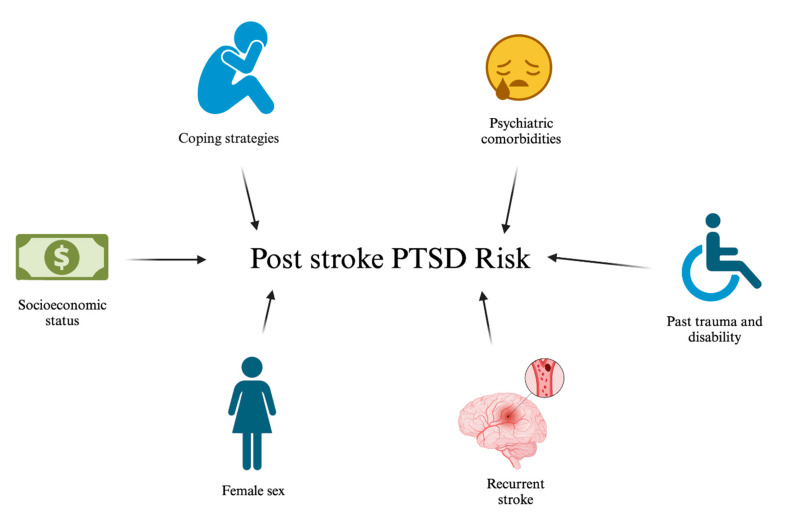
Post Stroke PTSD Risk. Several risk factors may be implicated in the development of post-stroke PTSD.

## Data Availability

Not applicable.

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
