# Peer review of "Neurologic Injury-Related Predisposing Factors of Post-Traumatic Stress Disorder: A Critical Examination"

_biomedicines, 2023, doi:10.3390/biomedicines11102732_

Round 1

Reviewer 1 Report

The topic of this descriptive review is interesting and important. There is no doubt about it.

However, now the manuscript is more like a chapter in a monograph, but not a descriptive review.

The strategy of finding publications for this review is not clear.

Major comments

The review needs modification, as 38 of the 101 cited publications are outdated (they were published more than 10 years ago). The "References" section needs a major revision (please use the template of this journal and the instructions for the authors).

Add a Discussion section and give a brief description of your main idea and your findings. Please explain how your review differs from previously published reviews. What's new? What are the limitations of your descriptive review?

The Conclusion section needs revision. It is not correct to give links to previously published articles here. Move these two large paragraphs to the Discussion section. Please briefly state in one or two sentences the main findings of your descriptive review.

Minor comments

Line 66 - Correct to "the sample”

Line 68 – Correct to “PTSD; 2) moderate”

Figure 1 - Add a note and explain all the abbreviations you used. The quality of the figure needs to be improved, since the fragment in the lower right corner is not readable.

Line 153 – Correct to “Traumatic Brain Injury”

Lines 190, 230 - Add a name for Figures 2 and 3

Double check the use of abbreviations:

-          - Do not use the abbreviation(s) if it (they) are used less than four times in the text of the manuscript.

-         -  If you have already explained the abbreviation when it was first used, then you do not need to explain this abbreviation again later in the text of the manuscript.

Author Response

Dear reviewer,

We appreciate your feedback and have made the following adjustments accordingly:

1) References - 5 reference changes to include more recent studies conducted addressing the same information discussed in the paper. Several older studies were still included for one of two reasons: a) No study has been conducted more recently with inconsistent results. b) The study provided important future directions that have still been unanswered. To make this apparent to readers, it was mentioned as a limitation in the discussion section.

2) A discussion section has been added summarizing findings (and their clinical significance) and discussing future directions.

3) A new conclusions sections was constructed clearly and concisely stating the key findings of the review.

4) All minor comments (line comments, figure changes, abbreviations, etc.) were addressed.

Thank you again for your contributions.

- Wiley Gillam

Reviewer 2 Report

The present review provides a compilation of recent evidence on the relevance of neurotrauma in different life cycle stages as a risk factor for PTSD and how the extent of an injury affects symptoms, pointing at different brain structures and neuromodulatory signaling as targets of interest in this respect. In order to  study this relation, three prevalent clinical scenarios have been addressed, namely, TBI, SAH and stroke. The authors also discuss the different limitations of the literature (sample size, criteria/version of DSM used, etc).
The structure of the review is established as general pathophysiological aspects, and then a specific section for each neurological trauma (TBI, SAH, stroke) with their respective graphical abstracts.

In the following parts, this reviewer would like to provide her opinion and suggestions constructively so that the authors can improve the quality of thei work.

The title current title “ Neurologic Injury and Post-Traumatic Stress Disorder” could be understood as simplistic as compared to the relevance of the new perspective that defines neurotrauma as a potential source of PSTD and its clinical implications in the screening of their symptoms in these three groups of patients. It is of critical clinical relevance that the strongest message/conclusion “These find-386 ings support the idea that many neurotrauma patients may benefit from regular PTSD 387 screening to prevent the worsening of symptoms if left untreated.” is given in a direct of indirect way both in the title and the abstract.

The abstract should be informative, a  summary report of the final conclusions, not the questions addressed in the current review. The key points of view clearly expressed in the conclusions (378-388) should also be part of the abstract. In my opinion: Contribution of internal medical events + relevance of screening for PTSD in neurotrama + limbic system foreseen as a target

Graphical abstracts: In this respect, figure 1, provides an illustration that can be confusing (should be understood without the need to read the legend), as ‘stress’ can be understood as due to other sources than ‘trauma’, so this ‘chronic or high acute’ and ‘during/after trauma’ should be included in the illustration. Similarly, the neuroanatomical structures (PFC, ACC, AMG, HC) or systems (Limbic and HPA axis) are indicated but only the functional alteration is indicated in some of them (Hippocampal ‘atrophy’ and ‘reduced activity’ in PFC and ACC).

Please, check this legend, as the quotation (1), (2), (3) does not follow a coherent estructure (location with respect to its respective sentence).

Line 187. Figure 2. A and B. The functional alterations can’t be read correctly due to the size. Also, please, reorganize the figure in a vertical presentation with normal functional scenario first (not last) in the top part,  and the TBI dysregulated scenario in the bottom part.

Line 228. Figure 3. Please, don’t use ‘stress’ without more specification as a term, but PSTD instead.

Sociodemographic factors are mentioned (i.e. lines 256, 304 etc), but a section to summarize the knowledge on how they modulate the clinical outputs is needed, in particular, with regards to gender medicine (differences or lack of differences) for neurotrauma, PSTD and their relationship (as done for stroke).

Protective factors should be also stated.

Figure 4: In coherence with previous figures, remove the frames around the subtopics.

Conclusions: Line 388, please, add the future directions as a new paragraphs and my suggestion is to be more extended on the guidance that you can provide.

It is loable that the present research was adressed without receiving external funding.

Minor

Please, check redundancies: i.e. line 10 correspondence

Author Response

Dear reviewer,

We appreciate your feedback and have made the following adjustments accordingly:

1) Title has been revised to "Neurologic Injury Predisposing Factors to Post-Traumatic Stress Disorder: A Critical Examination"

2) Abstract has been significantly modified to focus on the key findings of the review and their clinical relevance (rather than the questions the paper intended to answer).

3) Figure 1 - A different image for the HPA axis has been used, functional alteration is included for each step, all abbreviations have been explained in text, and (1), (2), (3) are now all implemented consistently in the text.

4) Figure 2 - Size adjusted to make text readable. Normal is first, TBI after.

5) Figure 3 - "PTSD" replaced "Stress" in graphic.

6) Sociodemographic factors and their impacts on clinical results have been added.

7) Figure 4 - Frames around graphical text removed.

8) Added a discussion section which summarizes findings (and clinical relevance) and discusses future directions.

9) New conclusions section which clearly and concisely states key findings.

Thank you again for your contributions.

- Wiley Gillam

Reviewer 3 Report

The paper titled "Neurologic Injury Predisposing Factors to Post-Traumatic Stress Disorder: A Critical Examination" presents a valuable exploration of the intricate relationship between neurologic injuries and the development of Post-Traumatic Stress Disorder (PTSD). The study is commendable for its comprehensive approach, focusing on three common neurologic injuries (traumatic brain injury, subarachnoid hemorrhage, and stroke) and their potential to lead to PTSD. However, there are areas where the paper could benefit from revision and further elaboration to enhance its overall quality and impact.

1. Methodological Details: This study would benefit from a section that delves into the methodology employed in the research. A clear description of how the studies were conducted, including data collection methods, sample sizes, and statistical analyses, will enhance the paper's credibility and allow readers to assess the quality of the research.

2. Discussion of Findings: While this study highlights the focus on CNS structures, limbic system associations, and the role of noradrenergic signaling, this study should delve deeper into these findings in the main body. Providing more extensive discussions and integrating relevant literature can help contextualize the results and support the paper's claims.

3. Clinical Implications: The current study mentions the importance of early diagnosis and treatment for PTSD in patients with neurologic injuries. Expanding on this aspect to discuss specific clinical implications, such as potential screening protocols or therapeutic interventions, would be highly valuable for healthcare professionals.

4. Future Directions: This study briefly touches upon areas that need further exploration, such as the impact of neurotrauma on the HPA axis. In the main body of the paper, you could dedicate a section to outlining potential future research directions, thereby encouraging further investigation into these critical areas.

5. Conclusion: This study should conclude with a concise summary of the key findings and their broader implications for both clinical practice and research. A well-structured conclusion will leave a lasting impact on the reader.

Author Response

Dear reviewer,

We appreciate your feedback and have made the following adjustments accordingly:

  1. Methodological Details - Although we did not go into explicit detail regarding all studies used in the paper, we decided to replace certain data and references with more updated findings where appropriate. We also mentioned in the discussion a potential limitation of the study as not defining rigid guidelines (particularly regarding date of publication) for which studies' results were discussed in the paper, so as not to be misleading to readers.
  2. Discussion of Findings - A discussion section has been added to the paper which goes into detail about the paper's key findings, clinical implications, and future directions.
  3. Clinical implications (recommended screening protocols) were added in the discussion section.
  4. Future directions were expanded on in the discussion section.
  5. A new conclusions section was constructed with clear, concise takeaways.

Thank you again for your contributions.

- Wiley Gillam

Round 2

Reviewer 1 Report

The authors modified the manuscript and improved the quality of the Figures, but not all the reviewer's comments were taken into account by them.

Line 159 - Do not use abbreviations in the subsection name.

Line 251 - Add a dot at the end of the Figure name.

Line 256, 268 - delete the abbreviation, you explained it earlier.

Line 345 - Do not use an abbreviation in the name of the Figure.

Double check the use of abbreviations. Please do not use abbreviations if they are mentioned in the text 4 or less times. Currently, abbreviations that are used once or twice have not been removed.

The Discussion section needs modification and expansion.

Author Response

Dear reviewer,

I have addressed all line comments. Regarding your first comment, I went back to your original suggestions and realized I had not properly responded to your comment about methods. In my expanded discussion, I added a potential limitation of the study as not utilizing a standardized protocol. I also mentioned how we worked to mitigate this limitation. The discussion has also been modified to go into further depth about this review's findings.

Thank you again for your feedback. It is greatly appreciated.

Very respectfully,

Wiley Gillam

Reviewer 2 Report

The authors have revisited their work in an accurate manner, and the revised version includes  changes (i.e. title, abstract) and new considerations (i.e. relevance of sociodemographic factors, conclusion remarks and future directions) that provide a better description of the  conclusions that this review can offer to the specialized audience but also understanding of their relevance in the clinical field. 

Author Response

Dear reviewer,

Thank you again for your feedback. It is greatly appreciated by our research team.

Very respectfully,

Wiley Gillam